Visual search characteristics of precise map reading by orienteers

Liu Yang liuyang0330@snnu.edu.cn
Department of Physical Education, Shaanxi Normal University , Xi’an, Shaanxi , China
Barnhart Anthony
Electronic publication date: 2019 Aug 23
Publication date: 2019
Volume: 7
Electronic Location ID: e7592
Received 2019 Feb 27; Accepted 2019 Jul 30
Copyright: © 2019 Liu
Copyright year: 2019
Copyright holder: Liu
License: This is an open access article distributed under the terms of the Creative Commons Attribution License, which permits unrestricted use, distribution, reproduction and adaptation in any medium and for any purpose provided that it is properly attributed. For attribution, the original author(s), title, publication source (PeerJ) and either DOI or URL of the article must be cited.
License URL: https://creativecommons.org/licenses/by/4.0/

Keywords: Orienteering, Precise map reading, Strategies, Visual attention

Funding: Humanistic and Social Science Foundation of Chinese Ministry of Education 16YJCZH063 This work was supported by the Humanistic and Social Science Foundation of Chinese Ministry of Education (No. 16YJCZH063). The funders had no role in study design, data collection and analysis, decision to publish, or preparation of the manuscript.

==============================
This article compares the differences in eye movements between orienteers of different skill levels on map information searches and explores the visual search patterns of orienteers during precise map reading so as to explore the cognitive characteristics of orienteers’ visual search. We recruited 44 orienteers at different skill levels (experts, advanced beginners, and novices), and recorded their behavioral responses and eye movement data when reading maps of different complexities. We found that the complexity of map (complex vs. simple) affects the quality of orienteers’ route planning during precise map reading. Specifically, when observing complex maps, orienteers of higher competency tend to have a better quality of route planning (i.e., a shorter route planning time, a longer gaze time, and a more concentrate distribution of gazes). Expert orienteers demonstrated obvious cognitive advantages in the ability to find key information. We also found that in the stage of route planning, expert orienteers and advanced beginners first pay attention to the checkpoint description table. The expert group extracted information faster, and their attention was more concentrated, whereas the novice group paid less attention to the checkpoint description table, and their gaze was scattered. We found that experts regarded the information in the checkpoint description table as the key to the problem and they give priority to this area in route decision making. These results advance our understanding of professional knowledge and problem solving in orienteering.

Introduction

Orienteering is a sport in which orienteers take a map and a compass and run to reach some points marked on a map as soon as possible (Eccles, Walsh & Ingledew, 2002a). It places high demands not only on orienteers’ physical fitness and strength but also on cognitive capabilities, such as attention, visual search, and decision-making. Route planning based on map information is an essential skill for orienteers. Therefore, map reading ability directly affects orienteers’ performance. There are dozens of directional map symbols, representing different landforms and terrain features. However, not all map symbols have navigational functions. Recognizing all the details on the map greatly reduces the speed of march in orienteering (Robertson et al., 1993). Therefore, the efficiency of map reading is important for orienteers to obtain the key information within a limited time (Muñoz-Nieto et al., 2013; Pesce et al., 2007). Therefore, the capability of visual searching can help orienteers quickly and efficiently recognize, process, and code the information on the map and to quickly and accurately find the key information in route planning (Henderson & Hollingworth, 1999; Henderson & Pierce, 2008). Eccles, Walsh & Ingledew (2002b) and other scholars applied the grounded theory in interviewing professional orienteers and put forward the cognitive theory of orienteering. From their research, we find that orienteers of expert competency can flexibly use a variety of skills to quickly identify map information and accurately extract pertinent information. A case study by Macquet, Eccles & Barraux (2012) points out that expert orienteer can simplify the navigational information to improve the efficiency of their route planning. For example, an orienteer will first select easily identifiable map information and quickly determine the checkpoints. As the training increases, orienteer’s ability of map reading is gradually developed and improved. Therefore, it is of great interest to examine the characteristics of map reading for orienteers at different levels, to summarize the differences in the visual attention of orienteers with different levels of experience in map reading, and to analyze the attention strategies of expert orienteers, in order to better select and train orienteers.

At present, most existing research focuses on the visual search feature of reasonable route selection between points on the map. Through interviews and questionnaires, it has been found that excellent orienteers choose obvious landforms and terrain features as “assistants” when planning a route (Macquet, Eccles & Barraux, 2012; Seiler, 1990). Eccles, Walsh & Ingledew (2002a, 2002b) summarized the visual search characteristics of expert orienteers by analyzing their self-statements about route planning and the experimental manipulation of the map board. It has been found that expert orienteers first focus on the information around the terminal point, whereas novice orienteers first focus on the information around the starting point. With the development of eye movement technology, the visual search strategy of orienteers can be measured more precisely as compared with the research methods of interview and self-report. Zhu et al. (2011) conducted eye-tracking research on competent players and novices, the results of which are consistent with those obtained from interviews (Eccles & Arsal, 2015). This research found that expert orienteers have a strong ability to simplify maps and focus only on the information related to the task. In previous research (Liu & He, 2018), the author systematically studied the general map reading of orienteers through eye movement technology and found that the map type and information quantity (i.e., map complexity) restricted the visual search and route planning of orienteers so that orienteers at different levels showed different visual search characteristics. Systematic evaluation of eye tracking shows that high-level orienteers have cognitive advantages in their visual search ability (Van Der Gijp et al., 2017). When reading simple maps, orienteers in the advanced beginner group and expert group tend to start searching from the beginning point, whereas when reading complex maps, orienteers in the expert group tend to start searching from the checkpoint description table.

In orienteering, it is necessary to investigate the visual search characteristics of precise map reading in which orienteers have approached a point and have to determine the exact location of point. Specifically, precise map reading requires orienteers to carefully identify and distinguish the information near a point and successfully reach the point (Fig. 1). In orienteering competition, the accuracy of the maps is also the key to successfully completing the route. In the present study, we adopt precise maps of different difficulties and measure the eye movement characteristics of the orienteer while searching for map information in order to explore the visual search characteristics of orienteers at different skill levels. The hypothesis is that experts, advanced beginners, and novices will be affected differently by the difficulty of the task during precise map reading and will have different visual search characteristics and processing features.

Figure 1 Example of a precise map for orienteering.

Methods

Participants

A total of 44 participants were invited to take part in the research, including 12 expert orienteers with an average age of 22.3 years, all of whom were active national orienteers at a skill level of class I or above, 16 advanced beginners with an average age of 21.2 years, all of whom were orienteers on a high-level university team at a skill level of class II or above, and 16 novice orienteers with an average age of 20.6 years, all of whom were students in an optional course of orienteering and had play orienteering for more than half a year. All participants had normal or corrected-to-normal visual acuity in their left and right eyes and were able to master the basics of orienteering. None had participated in similar experiments before. Informed consent was obtained from each participant. The study had been approved by the Ethics Committee of Shanxi Normal University.

Materials and design

A three (Competency level of orienteer: novice, advanced beginner, or expert) × two (map complexity: simple or complex) two-factor mixed experimental design was adopted for this research. Competency level of the orienteer was the between-subjects factor and the map complexity was the within-subject factor. The simple map was a park scene at a scale of 1:4,000 and the map information was composed mostly of landform features such as buildings. The complex map was a wilderness scene at a scale of 1:10,000 and the map information was mostly terrain features such as mountains. Three professors of orienteering were asked to rate the complexity of the selected maps on a 5-point scale from 1 ( very simple) to 5 (very complex). According to the ratings of these professors, 14 simple maps and 14 complex maps were selected (two maps were used as practice maps, and the remaining 12 maps were used as experimental materials). All experimental maps were based on actual scene maps used in a national orienteering competition and then redrawn by three national-level cartographers. The distance between points on all maps was two to three cm.

Apparatus

The Eyelink 1000 Plus eye tracker, with a sampling rate of 1,000 Hz, was used in the present experiment. The stimuli were displayed on a 17-inch CRT computer screen with a refresh rate of 85 Hz and a screen resolution of 1,024 × 768 pixels. The distance between participants’ eyes and the screen was approximately 60 cm. The laboratory was in a low-illumination environment. To reduce the effect of head movement on eye movement tracking, participants placed their jaws on the mandible holder at a proper height.

Procedure

The experiment required participants to read the map and plan route as if competing in a real competition. Before the formal experiment, the participants filled out a basic questionnaire prepared by the research team and the gender, age, training years, skill level, and other basic information about the participants was recorded. The experiment procedure is depicted in Fig. 2.

Figure 2 Flowchart for each experimental trial.

The experimental procedure was run by the E-Builder software provided by the SR Research Company. At the beginning of each trial, a small dot (drift correction) first appeared at the center of the screen. When the fixation of the test subject coincided with the small dot, the small dot disappeared and then the map was displayed. In the experiment, the subject was required first to read the presented map, and then plan a route they think was the best and remember it (stage of route planning). After the orienteer completed the planning, they were asked to press the “space bar” to make the map disappear. The route planning time was recorded by the computer. Next, the center of the screen presented a 200 ms prompt—“Please use your eyes to scan the route you have just planned.” And then the map was presented again. The orienteer was asked to quickly scan their planned route with their eyes on the map (stage of route scanning). After the scanning was completed, the orienteer would press the “space bar” and the map disappeared. There were four practice trails (two simple and two complex maps) before the formal experiment to ensure that the participants fully understood and were familiar with the experimental process. The formal experiment was completed in a total of 24 trials and the simple and complex maps were randomly presented in all trials.

Data processing

The original eye movement data was processed with a Data Viewer package provided by the SR Research Company. For eye movement data in the route planning stage, the researchers first removed the fixation point (i.e., the fixation identified by the eye tracker) outliers with fixation duration of less than 80 ms or more than 800 ms. The remaining data were analyzed through SPSS 17.0 statistical software package. The eye movement indicators included the number of fixations, the average fixation frequency, and the fixation track. Finally, the researchers organized each participant’s eye movement tracks for each trial stage into a video file in AVI format. Then we invited the three professors of orienteering to evaluate the quality of the route (i.e., fixation point track) the orienteer planed on a 5-point scale, and we used average of scores given by the professors as the indicator of quality of route planning. We focus on three eye movement indicators, that is, total fixation time, fixation frequency, and positions of the first five fixation points. The total fixation time refers to the total duration of all fixations in the stage of route planning. The fixation frequency refers to the number of fixation per second in the stage of route planning. The distribution of the first five fixation points refers to the positions of the first five fixation points on the map in the stage of route planning. We also showed the Heat map to show the distribution of the fixations in the stage of route planning.

Results

Behavioral indicators

Comparison of quality of route planning

For each trial, the participant’s eye movement track in the stage of route scanning was stored in a video file and three professors of orienteering were asked to evaluate the quality of the planned route by each orienteer on a 5-point scale. The average score of the evaluation was showed in Table 1. A 3 (Orienteer level: novice, advanced beginner, or expert) × 2 (map complexity: simple or complex) two-factor analysis of variance (ANOVA) found that the effect of orienteer level (F(2, 41) = 15.991, p < 0.001, η2 = 0.438) and map complexity (F(1, 41) = 26.198, p < 0.001, η2 = 0.390) were both significant, and the two-way interaction was also significant (F(2, 41) = 4.098, p < 0.05, η2 = 0.167). Simple effect analyses showed that under the simple map condition, the main effect of the orienteer level was not significant (F(2, 41) = 1.528, p > 0.05, η2 = 0.069), whereas under the complex map condition, the effect of orienteer level was significant (F(2, 41) = 16.219, p < 0.001, η2 = 0.442). The post hoc multiple comparisons for the effect of orienteer level under complex map condition showed that the quality of route planning in the novice group was significantly lower than that of the advanced beginner group (t = 3.75, p < 0.001) and the expert group (t = 5.56, p < 0.001), and that of the advanced beginner group was significantly lower than that of the expert group (t = 2.09, p < 0.05) (see Fig. 3).

Table 1 Route planning accuracy of orienteers.

Skill level	Map complexity	Route planning accuracy score	
		M	SD	
Novice	Simple	4.65	0.38	
	Complex	3.75	0.47	
Advanced beginner	Simple	4.74	0.39	
	Complex	4.31	0.36	
Expert	Simple	4.90	0.29	
	Complex	4.64	0.41	

Figure 3 Comparison of route planning accuracy of orienteers at different skill levels.

Comparison of route planning time

A similar 3 × 2 two-factor ANOVA found that the effect of orienteer level (F(2, 41) = 8.387, p < 0.01, η2 = 0.290) and map complexity (F(1, 41) = 6.134, p < 0.05, η2 = 0.130) were both significant, and the two-way interaction was also significant (F(2, 41) = 3.369, p < 0.05, η2 = 0.141). Simple effect analyses showed that under the simple map condition, the effect of orienteer level was not significant (F(2, 41) = 2.120, p > 0.05, η2 = 0.094), but under the complex map condition, the effect of orienteer level was significant (F(2, 41) = 10.226, p < 0.001, η2 = 0.333). The post hoc for the effect of orienteer level under complex map condition showed that the route planning time of the novice group was significantly longer than that of the advanced beginner group (t = 3.12, p < 0.05) and the expert group (t = 4.35, p < 0.001), whereas the difference between the advanced beginner group and the expert group was not significant (ps > 0.05) (see Table 2; Fig. 4).

Table 2 Route planning time of orienteers.

Skill level	Map complexity	Planning time (ms)	
		M	SD	
Novice	Simple	10,177.65	4,326.63	
	Complex	13,534.0	4,225.48	
Advanced beginner	Simple	8,870.62	1,982.12	
	Complex	9,721.80	2,985.79	
Expert	Simple	7,863.60	1,508.87	
	Complex	7,808.25	2,808.70	

Figure 4 Comparison of route planning time of orienteers at different skill levels.

Eye movement indicators

Total fixation time

The 3 × 2 two-factor ANOVA indicates that both the effect of orienteer level (F(2, 41) = 5.047, p < 0.05, η2 = 0.198) and that of map complexity (F(1, 41) = 4.794, p < 0.05, η2 = 0.105) were significant, as was the interaction between the two (F(2, 41) = 6.437, p < 0.01, η2 = 0.239). Simple effect analyses indicated that the effect of orienteer level was not significant in the simple map condition (F(2, 41) = 7.819, p < 0.05, η2 = 0.276) but was significant in complex maps (F(2, 41) = 7.819, p < 0.05, η2 = 0.276). The post hoc for the effect of orienteer level in the complex map condition found that the expert group showed longer total fixation times than the advanced beginner group (t = 2.51, p < 0.05) and novice group (t = 3.91, p < 0.001) and that there was no significant difference between the total fixation times in the advanced beginner group and in the novice group (ps > 0.05) (see Table 3; Fig. 5).

Table 3 Total gaze times in route planning.

Skill level	Map complexity	Gaze time (ms)	
		M	SD	
Novice	Simple	39.1	8.7	
	Complex	33.9	4.2	
Advanced beginner	Simple	37.4	9.7	
	Complex	42.5	14.4	
Expert	Simple	41.1	10.6	
	Complex	55.1	18.4	

Figure 5 Gaze times of orienteers at different skill levels.

Fixation frequency

The 3 × 2 two-factor ANOVA indicates that the effect of orienteer level (F(2, 41) = 0.256, p > 0.05, η2 = 0.012), the effect of map complexity (F(1, 41) = 0.148, p > 0.05, η2 = 0.004) and the two-way interaction (F(2, 41) = 0.364, p > 0.05, η2 = 0.017) were all nonsignificant. The results suggest that there was no difference in the fixation frequency when orienteers of different skill levels read maps (see Table 4; Fig. 6).

Table 4 Average gaze frequency in route planning.

Skill Level	Map complexity	Frequency of gaze	
		M	SD	
Novice	Simple	4.49	1.79	
	Complex	4.78	3.12	
Advanced beginner	Simple	4.34	1.41	
	Complex	4.72	1.96	
Expert	Simple	5.13	1.44	
	Complex	4.84	1.76	

Figure 6 Gaze frequency of orienteers at different skill levels.

Distribution of the first five fixation points

We compared the differences in the features of visual search via inspecting the distribution of the first five fixation points between orienteers at different skill levels. The first five fixation points of the novice group all fall between the starting point and the terminal point. The first five fixation points in the advanced beginner group fall on the checkpoint description table. The first five fixation points in the expert group fall on the checkpoint description table and the terminal point. As shown in Fig. 7, the expert group and advanced beginner group start from the checkpoint description table, but the expert group shows fewer fixation points than the advanced beginner group, which demonstrated a higher map reading efficiency; the novice group starts to search among spots in the map, with a special focus on the information between the starting point and the terminal point.

Figure 7 Movement of gaze of orienteers at different skill levels.

(A) Novice group. (B) Advanced beginner group. (C) Expert group.

Heat map of fixation distribution

Figure 8 gives three typical examples of fixation distribution in a map of a given complexity. Spots at which orienteers of different skill levels fixated are highlighted in green, yellow, and red. The green–yellow–red color scheme is used to represent the frequency of fixation, with green indicating the lowest frequency, yellow the intermediate frequency, and red the highest frequency. Through a comprehensive analysis of the fixation point distribution heat maps of orienteers at different skill levels, it can be seen that the more capable the orienteers are, the smaller the area they are likely to search. The red spots of the novice, advanced beginner, and expert groups are concentrated mostly at the terminal point. In terms of the duration of fixation, the expert group spends less time on the checkpoint description table than the advanced beginner group, whereas the novice group spends none, indicating that the expert group has hgher map reading efficiency.

Figure 8 Attention distribution of orienteers at different skill levels.

(A) Novice group. (B) Advanced beginner group. (C) Expert group.

Discussion

Analysis of orienteers’ route planning accuracy

A detailed analysis of the quality of route planning shows that novice, advanced beginner, and expert groups all show less accuracy in route planning in complex maps than in simple maps. This means that the performance of orienteers at different skill levels is subject to the complexity of the maps, which was consistent with what has been discovered in sketch map reading studies (Liu & He, 2018; Zhu et al., 2011). Visual search generally needs to be done through top-down target attention and feature identification (Wolfe & Horowitz, 2017). When the cognitive demand in a scenario increases, the limitations of visual attention processing will increase the time needed to make decisions (Zhou & Liu, 2010). In simple precise map reading scenarios, cartographic information is basically about houses. In complex maps with brown contours as geomorphic indicators, more information is included, posing more challenges to orienteers as they must identify a lot of information carefully; this leads to much more effortful information processing and slows down the route planning speed.

Prior research demonstrated that long-term training can speed up and improve decision making (Liu & He, 2016). As for orienteering, providing long-term training in knowledge and skills will better regulate the structure and sequence of information processing. Meanwhile, the ability of low-level orienteers to identify key information is subject to their working memory capacity (Eccles, Walsh & Ingledew, 2002a). Therefore, novice orienteers are more likely to be influenced by the complexity of maps, which results in a worse performance than that in the expert group.

Analysis of orienteers’ visual search efficiency

Visual search efficiency is determined by route planning time and fixation frequency. Route planning time shows how much information is processed by the orienteer; the longer it takes, the less efficient information processing is thought to be. The fixation frequency refers to the number of fixation per unit time, which can be regarded as the processing speed of the map search; the more capable the orienteer is, the more information he or she can obtain and store, and the more efficient information processing is; in other words, more information is obtained per unit time (Wiley & Jarosz, 2012).

According to our findings, in complex maps the advanced beginner group and expert group spent less time on route planning than the novice group, which indicates that expert orienteers are better at gathering and processing information and thus have higher visual search efficiency (Ericsson & Lehmann, 1996). Eye-tracking data for orienteers’ precise map reading reveals that task difficulty can differentiate the total fixation times between the expert, advanced beginner, and novice groups. Excellent orienteers can gather useful information using an effective fixation and ignore useless information, saving more time for important zones.

This characteristic likely develops because expert orienteers have acquired sophisticated skills over time and may be better at visual search (Eccles, Walsh & Ingledew, 2006). For example, their experience can help reduce the cognitive load in information processing so that they can better search for information quickly (Williams, 2000), encode valuable information efficiently (Baber & Butler, 2012; Eccles, 2008), and flexibly adopt various features (Eccles & Arsal, 2015). Novice orienteers are less efficient in map searching and cannot quickly identify valuable information. Therefore, they do not have an advantage in the cognitive processing of specific information.

It takes three-dimensional thinking to understand cartographic signs on complex contour maps. It is more difficult to identify cartographic signs on complex maps than on simple maps because of dimensionality. Our eye movement data revealed that map complexity differentiates the visual search efficiency of the orienteer. Complex maps pose more challenges to the visual search efficiency of the orienteers. With the growing complexity of maps, more cognitive effort is required from the orienteer that occupies a large proportion of the cognitive capacity and they need to store and retrieve information to and from memory, leading to a longer duration for information retrieval and identification (Rinck et al., 2003).

Analysis of orienteers’ visual search patterns

A visual search pattern is the way orienteers search for useful information. More specifically, it refers to the trajectory of the fixation formed by the connection of each fixation point, as well as the heat map of the fixation, both of which exhibit the search characteristics of the orienteer. During precise map reading, subjects’ visual search patterns display identical features as those in sketch map reading (Eccles, Walsh & Ingledew, 2006; Liu & He, 2018; Zhu et al., 2011). The higher the skill level of the orienteer, the smaller the area the fixation will cover. Fixation distribution become more concentrated as their orienteering skills improve. Skilled orienteers tend to pay more attention to the changeable information, because they can efficiently shift or allocate their attention to reduce the cognitive load for processing information so as to resist to the constraints (Joseph et al., 2009). The distribution of novice orienteers’ fixations was pretty scattered. Another important finding of this study is the difference in visual search of the checkpoint description table between orienteers of different skill levels. Through the trajectory of the first five fixation points, we found that the expert group starts at the checkpoint description table so that they can gather the point information and then extract key information around the points. Advanced beginners show a similar pattern of visual search, but they produce more fixations than experts, which indicates that there are differences between orienteers of the two skill levels in terms of familiarity and understanding of the checkpoint description table. The novice group fixates back and forth between the starting point and the terminal point, leading to a scattered, deconcentrated fixation trajectory; the first five fixation points are not on the checkpoint description table either. Heat maps of fixation distribution also indicate the importance of the checkpoint description table, in which we find that the expert group’s fixations cover more areas of the map than those of the advanced beginner group despite their fixation distribution to the checkpoint description table. This was an indication of swift information retrieval by expert orienteers. The novice group gives no attention to the checkpoint description table. The checkpoint description table is a detailed illustration of where the points are and defines their features. This was a useful guide for orienteers to seek the best routes as quickly as possible. Gasser (2016) pointed out that the checkpoint description table is crucial for orienteers’ performance in orienteering. The study also finds that experts outperform orienteers of other skill levels in map reading around the checkpoint description table with a higher efficiency of visual search.

To maximize the potential of the orienteer, the selection and training of orienteers are both important (Newton & Holmes, 2017). This finding reminds us that we should increase the familiarity of the orienteer with the map information during training. More legends and information are contained in complex maps, and thus novice orienteers are not able to understand the information. Therefore, it is necessary to train them to be familiar with the map so as to gain more experience and understand the information more quickly. Meanwhile, it is necessary to train them in identifying and attending to the checkpoint description table. Advanced beginners’ understanding of information in the checkpoint description table should be increased as well so they can effectively gather the information in the checkpoint description table. Previous studies have shown that teaching novices specific cognitive processing skills can improve their orienteering skills, such as positioning and using specific clues in a certain environment (Singer et al., 1994). Therefore, it is feasible to design an efficient route toward the terminal point by training novice orienteers to pay attention to the checkpoint description table.

This study also has a limitation. It was carried out in laboratory, so its ecological validity needs further examination. Participants reported that their route planning in the experiment was similar to, yet still different from, real-life situations. For example, studies have revealed that pressure will influence orienteers’ performances of memory (Robazza et al., 2018) and attention (Eccles, Walsh & Ingledew, 2006) in map reading in running. Moreover, attention shifts between the map and the environment (Mottet, Eccles & Saury, 2016; Pesce et al., 2007) will influence the visual search patterns of novice and expert orienteers and the result of the competition. Therefore, time and space can be introduced in future studies to improve their authenticity and thus improve ecological validity.

Conclusions

The present study shows that the complexity of a map influences orienteers’ route planning in precise map reading. In complex maps, the higher skill level of the orienteers, the higher score they can get on route planning and the more efficient their visual search. The novice group spends more time on route planning than the expert and advanced beginner groups and expert orienteers show more cognitive strength in seeking key information. The checkpoint description table plays a guiding role in precise map reading. Experts and advanced beginners both pay attention to the checkpoint form, but the former understands information faster and are more concentrated in searching information, whereas novices pay no attention. This study provides sound information for designing precise map reading training sessions for novices, which may then improve their performance in competitions.

Supplemental Information

Supplemental Information 1 Raw data 1.

Original data shows that: the accuracy of the route planning by the three directional movement project expert control subjects fixation point trajectory 5 scores for accuracy, measurement of each subject’s average score.

Click here for additional data file.

Supplemental Information 2 Raw data 2.

Using the SR Research Data provided by the company viewer Data analysis program for Data processing, accurate knowledge of different horizontal directional players figure route planning time, at times, fixation frequency Data acquisition and analysis.

Click here for additional data file.

Additional Information and Declarations

Competing Interests

Author Contributions

Human Ethics

Data Availability

The authors declare that they have no competing interests.

Yang Liu conceived and designed the experiments, performed the experiments, analyzed the data, contributed reagents/materials/analysis tools, prepared figures and/or tables, authored or reviewed drafts of the paper, approved the final draft.

The following information was supplied relating to ethical approvals (i.e., approving body and any reference numbers):

Shaanxi Normal University granted Ethical approval to carry out the study within its facilities.

The following information was supplied regarding data availability:

The raw measurements and raw data are available in the Supplemental Files.

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
