# Peer review of "Visual search characteristics of precise map reading by orienteers"

_PeerJ, doi:10.7717/peerj.7592_

## Round 0.1 · original submission · Major Revisions

I have received two reviews from experts in the field (please see below). Both agree that your manuscript is in need of substantial revision. While the paper could make an interesting contribution to the literature, the current document lacks clarity both in style and substance. The writing style makes the narrative difficult to follow, and it is quite difficult to conceptualize the experiment without more coherent operational definitions of the dependent variables.

I encourage you to revise your manuscript, at which point I will once again seek the advice of these reviewers to evaluate whether their critiques have been addressed.

I request that you add a statement to the paper confirming whether, for all experiments, you have reported all measures, conditions, data exclusions, and how you determined your sample sizes. You should, of course, add any additional text to ensure the statement is accurate. This is the standard reviewer disclosure request endorsed by the Center for Open Science [see http://osf.io/project/hadz3]. I include it in every review.

Best,
Anthony Barnhart

[]

Reviewer 1 ·

Basic reporting

Major points
While the paper might offer some interesting insights into the perceptual-cognitive components of skilled performance in orienteering, and as such into skilled performance more generally, the presentation of the paper falls well short of the standards expected of a scientific paper. While some of these issues are superficial, many are not, which means that the paper in its current form is difficult to follow, and has methodological and conceptual issues. Much more attention must be given to the format, writing, and citation and referencing, which are poor. One example of these shortcomings is that several citations provided in support of an assertion appear to be entirely unrelated to the assertion (I provide a few specific instances below). More attention to detail is needed in the methods section; see my comment below about how there is a proposal in the measures section that a retrospective report was elicited from the participants but this method is not mentioned again and no associated data are presented. There also appear to be key conceptual issues; I detail one such problem below concerning the distinction made between general and precise map reading. My recommendation is to bring on board to the research team a more experienced researcher to critique all aspects of the current paper in detail and aid with a re-presentation of the paper and also a native English to enhance the writing, which will aid reader comprehension.

Minor points
Line 33: A critical missing piece of information here is that the orienteer must visit these points on foot
Line 35: I am quite sure strength is not important to orienteering and that measures of intelligence are at best weak predictors of orienteering performance.; see the literature on prediction of sport performance using domain-general abilities (e.g., intelligence) (generally poor) versus domain-specific perceptual-cognitive skills (generally good)
Lines 43 and 47: I cannot see how the Robertson et al. and Happe and Frith citations can possibly support the claims made prior to them. The Happe paper concerns autism, not orienteering. I found other instances of inappropriate support in this regard.
Lines 63-68: I’m not sure I understand the distinction made between general map reading, requiring athletes to plan a running route between points, and precise map reading, requiring athletes to combine information checklists and clarify the running route to find target points. Sometimes a distinction is made in the sport of orienteering between fine navigation, which involves very frequent comparisons between the map and surrounding environment to be sure that one knows where one is located in the world given the information on the map, and less fine navigation, where less frequent map-world comparisons are made. Fine navigation is usually required because the cost, in terms of time, of making a navigational error is higher than normal, and two instances where this is true are in complex terrain, which involves few distinguishable features, and when nearing a control (referred to as a target here) within a given leg; often, this is in the last 50-100 metres of a leg. I am not sure I see that the authors’ description of precise map reading is the same as my understanding of fine navigation. While the description of precise map reading mentions finding the target, what I see in the associated images is some less complex (Fig 1) and more complex (Fig 2) terrain. In other words, I see a conflating of approaching the control with complex terrain; yes, they both require fine navigation but for different reasons. And I don’t agree that less fine (general, in the authors’ terms) map reading does not require combining information checklists. If you orienteer in relatively easy terrain and don’t need to engage in much fine/precise map reading and navigation, you are still likely to consult the control checklist on the map when approaching a control to know where the control is located.
Line 54: Manzanares etc. This is not how to provide a citation.
Line 81: The citation is Arsal, 2015 but the reference, which is correct, is Eccles and Arsal.
Line 82: “Liu yang, 2018” Citations ask for the last name only; no first names
Line 87: “Van” citation; I don’t think this can be correct
Line 158: The reader is told that the participants completed a retrospective report after the experiment. I assume this means a retrospective verbal report of thoughts recalled from during the task, but no details or supporting literature (e.g., Ericsson & Simon) are provided. Furthermore, the method is not mentioned again and no results are provided.
Line 164: “Total planning time”. The variable is total gaze time, surely. I think you might be inferring that total gaze time reflects total planning time. This is an important distinction.

Experimental design

See above

Validity of the findings

See above

Additional comments

See above

Reviewer 2 ·

Basic reporting

This article reports information on an interesting study design. The authors are interested in interactions of expertise level (novice, intermediate, and expert) and map complexity (simple, complex) on accuracy and response times for reading and interpreting topographical maps, as well as for route planning using topographical maps. Eye tracking was used to record participants' eye movement data and the eye tracking data was analyzed to assess for the hypothesized interactions. The authors begin with an introduction to the sport of orienteering and do a good job of explaining the variables of interest within orienteering. A sufficient review of the literature is also present in the introduction. The authors state their hypothesis at the end of the introduction.

The language used by the authors is at times very difficult to follow. This appears to be the result of 1) poor sentence construction 2) nebulous use of terminology. For an example of poor sentence construction, please refer to the sentence in line 54 through line 57: "In addition, Manzanares etc (2017) empirical research, It further shows that athletes with different skill levels will have different visual behaviors, and the top ranked players can adopt a more active visual search strategy, and more experienced athletes can better obtain favorable information from important positions." Sentences such as these pose the reader undue challenges in deconstructing what the authors are attempting to communicate.

For an example of nebulous terminology, please refer to the hypothesis (lines 104 to 107). The hypothesis states that different levels of expertise will be "affected differently by the difficulty of the task during precise map reading and will have different visual search characteristics and processing strategies." It is unclear, however, exactly what is meant by "visual search characteristics and processing strategies." Furthermore, in their results section the authors use terminology specific to eye tracking which is not well defined. What is meant by "response times" (line 190), "gaze times" (line 204), and "average gaze frequency" (line 216)? This lack of specific explanations for what each of these variables mean with respect to the authors' hypothesis and results make it extremely difficult if not impossible to reconstruct the effects being explained in the results and discussion sections.

In looking at the raw data, the same problem exists. The raw data includes columns for "total gaze times", "gaze times", and "average gaze times" however it is unclear what any of these column headings mean with respect to their values. I was unsuccessful in reconstructing what any of the column headings refer to with respect to the data below the column headings or how the column headings related to each other.

The authors discuss an analysis of route-planning accuracy (lines 248-265), however no raw data is provided. The authors refer to the use of three orienteering expert trainers who scored "the correctness of each participant's gaze point track on a 5-point scale and then give a final score based on the average of the experts' scores, please include the raw data for this analysis.

Experimental design

As stated above, there are serious problems with syntax and lack of specific explanations which seriously impede the reader's ability to understand and interpret the hypothesis and results section.
Some of the methods described by the authors are presented in a much more understandable and clear fashion. For example, the experimental process (lines 139 to 158) was written in a very clear manner which allows sufficient detail to replicate this process.

Validity of the findings

As stated in section 1 ("Basic Reporting"), the results section (lines 170 - 244) is very difficult to follow. This difficulty primarily arises from a lack of clear and explicit descriptions of terms such as "response times", "gaze times", and "average gaze frequency." It is strongly recommended that the authors revisit the use of these terms in this article to provide clearer and more explicit descriptions. This will aid greatly in readers' ability to more clearly understand the hypothesis and stated findings of significance.

---

## Round 0.2 · Minor Revisions

The reviewers from round 1 have re-examined your revision and both are quite positive about the manuscript. However, they have both noted details that need to be addressed before the piece meets threshold for publication. All of these items are issues of clarity that should be rather easily addressed. I look forward to reading a revised version of this manuscript.

Reviewer 1 ·

Basic reporting

See below

Experimental design

See below

Validity of the findings

See below

Additional comments

The paper is much improved such that I can now understand it much more easily when I read it, and it is an interesting study that is quite well-designed and will contribute to our general understanding in the area of expertise research in psychology. Specifically, there is some nice use of cognitive process tracing and performance measures to uncover classic expertise by problem complexity interactions: that is, experts show their superiority over novices more clearly when the problems are complex enough to require that they draw on their more elaborate and refined problem representations. However, there are still some outstanding issues that need to be addressed, but these are now mainly superficial. That said, they must be addressed if the reader is going to be able to follow the work; readers won’t read all the work and thus won’t use and in turn cite the work to inform their own research if they can’t understand the work.
The authors refer to route planning “correctness” and “accuracy” but I think it is unlikely that there can ever be a single optimal route between two controls. My feeling is that the three expert judges can certainly judge the quality of a route, seeing whether one route is a better choice than another, and so I suggest using the word planning quality rather than planning correctness or planning accuracy. Check for every instance throughout the paper and make this change.
When follow up tests are run (e.g., line 190), make sure to state which level of the factor these are run on. On 190, the post-tests are presumably run on the complex level of the factor only because the main analysis had showed a main effect of skill level at the complex level but not the simple level; currently, this is not stated explicitly. Check all instances throughout the paper and correct this.
Make clear that all of the eye tracking data analysed are based only on data from the route planning stage of the study. This is important because the eye was tracked in the trail stage to identify the route planned, so the reader might be unsure (this reader was unsure initially) about whether the data from the trial stage were also analysed in these analyses.
Line 211: Some statistics are missing here.
Line 218: “…and the number of the total gaze is 90%”. Total gaze directed to the checkpoint? Is terminal point the same as checkpoint? If so, use one term only. This paragraph is unclear generally and needs to be very carefully revised if the reader is going to be able to understand what’s being presented.
Figures 7 and 8 is uninterpretable. Remove these.
Check all tables and figures. Units of measurement should be clearly provided for all. For accuracy scores, provide a note against tables and figures specifying the scale range and anchor points so the reader can interpret what the values mean; without these, the values are meaningless.
The efficiency in gaze behaviors observed by the experts in this study is consistent with the notion of experts’ circumvention of processing limitations generally (I suggest citing: Ericsson, K. A., & Lehmann, A. C. (1996) Expert and exceptional performance: Evidence of maximal adaptation to task constraints. Annual Review of Psychology, 47, 273-305) and expert orienteers’ circumvention of visual attentional limitations specifically (i.e., the experts derive key information from environmental displays using minimal visual attentional resources). As such, I suggest citing: Eccles, D. W. (2008). Experts' circumvention of processing limitations: An example from the sport of orienteering. Military Psychology, 20, S103-S121.
The citations and references are extremely poorly presented and MUST be revised so that there are no errors at all. Carefully follow APA format so each reference adheres fully to this format.

Reviewer 2 ·

Basic reporting

* Line 21 forward: The authors use the terms "orienteer" and "athlete" interchangeably. These terms are not synonymous. Orienteering is a specialized type of athleticism and it is confusing to have the terms “orienteer” and “athlete” used interchangeably. It is especially confusing because levels of expertise for orienteer as a variable of interest whereas levels of expertise for general athleticism may have nothing to do with map reading.
* Unclear use of terminology in the following instances: lines 80-81 “positive thinking characteristics”; line 87 “the accuracy of maps”; beginning with line 155 forward, please define specifically how authors are operationalizing the term “gaze”, this extends to line 196 “gaze time”
* Lines 218-219 “and the number of the total gaze is 90%” is unclear
* Lines 219-220 “and the number of the table and the terminal point is 75% of the total gaze” What does this mean, exactly?
* Line 244 “The visual search of tasks generally needs to be done through top-down target attention and feature analysis” What is meant by “visual search of tasks”?
* Line 261-262 “The gaze frequency refers to the number of gaze per unit time, which can be regarded as the processing speed of the map search…” This explanation is unclear. Does gaze frequency mean the amount of time a participant looks at a certain area of the map?
* Line 301 Use of the term “checkpoint form” is unclear. Do the authors mean to refer to the use of the map legend, which contains explanations for symbols used in the map?

Experimental design

No comment

Validity of the findings

This is an interesting line of research with worthwhile results. I maintain that the authors would help to promote this study and future research by using a narrative that is more concrete and specific in the use of terminology and phrasing. Please see section of “Basic Reporting” for specific details.

Annotated reviews are not available for download in order to protect the identity of reviewers who chose to remain anonymous.

---

## Round 0.3 · Minor Revisions

I did not send your manuscript out to reviewers, as I am satisfied that all of the substantive critiques have been addressed. However, there are still many grammatical errors throughout the document that need to be fixed. I have attached a commented version of your document where I made grammatical changes and noted other changes that need to be made before this can be published.

In the prior draft of the manuscript, one reviewer noted that it is confusing to use "orienteer" and "athlete" interchangeably. I have still noted instances where you use the word "athlete." I ask that you change these instances to be "orienteer." I've also noted some instances where it makes sense for you to refer to "fixations" rather than "gazes." I think you almost always mean to reference fixations, rather than gazes.

I look forward to reading one further revision of this manuscript.

---

## Round 0.4 · accepted · Accept

Thank you for making these changes. I am happy to accept your manuscript for publication in PeerJ.